# First evaluation of an absolute quantum gravimeter (AQG#B01) for future field experiments

Anne-Karin Cooke[1], Cédric Champollion[1], and Nicolas Le Moigne[1]

[1]Géosciences Montpellier, Univ. Montpellier, CNRS, Univ. des Antilles, Montpellier, France

**Correspondence:** Anne-Karin Cooke (annekacooke@gmail.com)

**Abstract.** Quantum gravimeters are a promising new development allowing for continuous absolute gravity monitoring while remaining user-friendly and transportable. In this study, we present experiments carried out to assess the capacity of the AQG#B01 in view of future deployment as a field gravimeter for hydro-geophysical applications. The AQG#B01 is the field version follow-up of the AQG#A01 portable absolute quantum gravimeter developed by the French quantum sensor company Muquans. We assess the instrument's performance in terms of stability (absence of instrumental drift) and sensitivity in relation to other gravimeters. No significant instrumental drift was observed over several weeks of measurement. We discuss the observations concerning the accuracy of the AQG#B01 in comparison with a state-of-the-art absolute gravimeter (Micro-g-LaCoste, FG5#228). We report the repeatability to be better than 50 nm s$^{-2}$. This study furthermore investigates whether changes of instrument tilt and external temperature and combination of both, which are likely to occur during field campaigns, influence the measurement of gravitational attraction. We repeatedly tested external temperatures between 20 and 30 °C and did not find any significant effect. As an example of a geophysical signal, a 100 nm s$^{-2}$ gravity change is detected with the AQG#B01 after a rainfall event at the Larzac geodetic observatory (Southern France). The data agreed with the gravity changes measured with a superconducting relative gravimeter (GWR, iGrav#002) and the expected gravity change simulated as an infinite Bouguer slab approximation. We report two weeks of stable operation under semi-terrain conditions in a garage without temperature-control. We close with operational recommendations for potential users and discuss specific possible future field applications. While not claiming completeness, we nevertheless present the first characterisation of a quantum gravimeter carried out by future users. Selected criteria for the assessment of its suitability in field applications have been investigated and are complemented with a discussion of further necessary experiments.

## 1 Introduction

Gravimetry studies the spatial and temporal variations of the Earth's gravity field which can be linked to changes in mass distribution studied in various disciplines of the Earth sciences. Applications include geodesy and large-scale geodynamics such as tectonics and slow subsidence (Camp et al. (2011); Hwang et al. (2010)) as well as crust deformation and glacial isostatic uplift (Mazzotti et al. (2011); Olsson et al. (2019)). Gravimetry has furthermore proved to be a tool in natural hazard assessment such as monitoring of volcanic activity (Bonvalot et al. (1998), Carbone et al. (2017)) or mapping of underground voids or the study of earthquakes (Imanishi (2004)). Applications in energy and resources comprise geothermal fields (Pearson-

Grant et al. (2018)), $CO_2$ storage reservoirs (Sugihara et al. (2017)), or artificial groundwater recharge facilities (Kennedy et al. (2016)). Gravimetric methods find furthermore application in the context of oil and mineral exploration and prospecting (e.g. Ferguson et al. (2007); Hinze et al. (2013)). In hydrology gravity measurement provide possibilities to monitor storage dynamics of local and landscape-scale groundwater resources (e.g. Creutzfeldt et al. (2008); Creutzfeldt et al. (2010); Jacob et al. (2010); Hector et al. (2014); Hector et al. (2015); Fores et al. (2016a); Güntner et al. (2017)) and even evapotranspiration rates (Van Camp et al. (2016)). Applications and active gravity research fields have been extensively reviewed by Crossley et al. (2013) and Van Camp et al. (2017).

Gravimeters are devices that measure the gravitational attraction $g$ on the Earth's surface. Nowadays gravimeters based on different measurement principles exist for various applications. Gravimeters can be characterised by their measurement performance: the repeatability of a measurement refers to the agreement between repeated measurements and is usually assessed by carrying out several repeated location changes in between measurements. The sensitivity (or precision) of a gravimeter is a relative uncertainty and refers to the smallest change in gravitational acceleration that the gravimeter is able to detect. We refer to stability as the absence of a significant instrumental drift in time or correlated noise. The accuracy of a gravity measurement describes to which extent it can be considered as correct in absolute terms and refers to the uncertainty of a measurement relative to an absolute standard (Niebauer (2015)).

Relative gravimeters sense the gravitational attraction indirectly by measuring the force needed to counteract gravitation by stabilizing a test mass and are used to monitor relative gravity changes. These devices show drifts that can become important within days (spring gravimeters) or months (superconducting gravimeters) (Van Camp et al. (2017)). Relative gravimeters require regular calibration with absolute gravimeters as reference stations and repeated looped or absolute measurements to eliminate drifts in field surveys (Hector and Hinderer (2016); Kennedy and Ferré (2015)) and can be sensitive to temperature changes (for example with a relative spring gravimeter as in Fores et al. (2016b)). The most sensitive relative gravimeters are superconducting gravimeters and achieve a high precision of about $0.1 \, \mathrm{nm \, s^{-2}}$ while measuring continuously at a sampling rate of 1 Hz. They are based on magnetic levitation instead of a mechanical spring (Hinderer et al. (2015)).

Absolute gravimeters estimate the norm of the gravitational acceleration $g$ during vertical free fall in vacuum (Niebauer et al. (1995)). Absolute gravimeters are mostly based on a free-fall corner-cube retro-reflector in a vacuum chamber with an instrumental uncertainty in the order of a few tens of $\mathrm{nm \, s^{-2}}$ (Niebauer (2015)). Currently available absolute gravimeters are not suitable for continuous monitoring due to mechanical parts with a limited lifespan. This limits the number of free fall experiments and requires frequent instrument repairs. Their operation usually requires a high technical skill level and they can not be operated by a non-specialist.

Quantum sensing offers new possibilities for measuring inertia and the development of quantum absolute gravimeters. The general measurement principle of an absolute quantum gravimeter (AQG) is that of matter-wave interferometry ((Peters et al. (2001)); Merlet et al. (2010)). The atoms can be exploited as test masses as well as a tool to measure the travelled path in order to sense gravity (Peters et al. (2001)). A laboratory realisation, the Cold Atom Gravimeter (CAG) developed at LNE-SYRTE in the context of the Kibble balance has demonstrated unprecedented performances both in sensitivity and accuracy. It has since participated at International Comparisons of Absolute Gravimeters (ICAG) showing a better short-term sensitivity than

absolute gravimeters and a well quantified accuracy budget (Jiang et al. (2012); Francis et al. (2013); Gillot et al. (2014)). Numerous research institutions and private companies work on different realisations of cold atom gravimeters (Geiger et al. (2020)) such as GAIN (Germany; Hauth et al. (2013)) or WAG-H5-1 (China; Huang et al. (2019)).

Robust atom manipulation, as a crucial step towards the realisation of a field instrument, was achieved by atom trapping and
detection without the necessity to align several optical components. This has been realised thanks to a single hollow pyramidal reflector, allowing operation with a single optical beam, and leading to a compact implementation (after Bodart et al. (2010)). An exhaustive review on the state-of-the-art of cold-atom gravity-inertial sensors, different sensor types, applications, and differences in performance has been provided by Geiger et al. (2020). The first commercially available gravimeter (AQG#A) based on this technology has been developed by Muquans. The compact design enabled a mobile instrument that does not
require special training in operation. Advances in on-the-fly correction of external effects have contributed to a compact and stable instrument of high sensitivity measuring at 2 Hz sampling rate. A stability better than $10\,\mathrm{nm\,s^{-2}}$ during one month of operation was observed, and the instrument's repeatability was preliminary quantified to be in the same order of magnitude (Ménoret et al. (2018)) for 24 h of averaging.

The main aims of this study are to assess the stability and repeatability of the first field absolute quantum gravimeter
(Muquans, AQG#B01) in view of future deployment as a field gravimeter for hydro-geophysical applications. The assessment is done during continuous measurements and experiments (impact of orientation or transportation) in comparison to absolute and relative gravimeters. The sensitivity to tilt and temperature changes as well as the interaction between both are of crucial importance to assess the suitability of the AQG#B01 as a field instrument. Its sensitivity to ambient temperature changes is evaluated by conducting tests in a controlled environment. Finally, recommendations are presented for the future use of the
AQG#B01 in field experiments.

## 2 Site and instrumental set-up

The field site allows for the monitoring of gravity with complementary instruments and of environmental variables to link gravity variations to mass changes occurring in the surroundings of the study site. Instrument tests and monitoring from December 2019 to April 2020 were carried out at the Larzac Observatory, which is part of the French National Research Infrastructure
OZCAR (Gaillardet et al. (2018); OZCAR-RI H+ Larzac - France) and the European long-term environmental monitoring network ELTER (Mollenhauer et al. (2018)). The Larzac Observatory is located on the La Jasse site in L'hospitalet-du-Larzac in Southern France. The observatory is highly instrumented with hydro-meteorological monitoring (eddy co-variance flux tower, rain gauges) and well suited for hydrogeophysical studies (Fores et al. (2018)). It is also part of the French seismological and geodetic network RESIF (Volcke et al. (2014)). The Larzac site further serves as a site for gravimeter testing, for instance in a
study on the AQG#A01 or gPhoneX (Micro-g LaCoste; Fores et al. (2019)). A superconducting gravimeter on the site (GWR, iGrav#002) has been monitoring gravity variations for almost a decade. Residual gravity changes caused by hydrological mass changes of less than $100\,\mathrm{nm\,s^{-2}}$ have been identified (Fores et al. (2016a)). An absolute free-fall gravimeter (Micro-g-LaCoste, FG5#228) has been transported to and operated at the site. During the International Key Comparison of Absolute Gravimeters

in 2017 (CCM.G-K2.2017), the FG5#228 showed a degree of equivalence of $3\,\mathrm{nm\,s^{-2}}$ with the 12 other absolute gravimeters and $55\,\mathrm{nm\,s^{-2}}$ uncertainty within 95% confidence (Wu et al. (2020)). In this study, the FG5#228 serves as a reference. In the Larzac observatory, the AQG#B01 and FG5#228 were operated on the same concrete pillar with approximately one meter distance between both instruments.

5    The AQG#B01 is the field version follow-on of the AQG#A01 described in Ménoret et al. (2018). It is based on the same measurement specifications and overall architecture but underwent a complete system redesign in order to meet outdoor operation requirements and to increase the stability of the measurement. The laser module and sensor head have been provided with an active thermal stabilization, allowing for a potential operation temperature range between 0°C and 40°C. Power consumption has been reduced to 250 Watt. Battery operation is possible but has not been tested yet. This study focuses on operation

10  relying on external power supply. Improvement in ease of use and transportability has been achieved with each element weighing 40 kg or less. The sensor head, the laser system and the control unit come each with a dedicated transport box that can be carried by two people. A fourth transport box is provided for the tripod, cables, connectors and the laptop. The enclosure for the lasers and the sensor head has been made water- and airtight. Connectors and cables are suitable for field conditions and a reduction in the number of connectors further facilitates fast and efficient field set-up. As an example, shifting the AQG#B01

15  in the Larzac observatory from one measurement location to another takes around 5 minutes for one person. The AQG GUI software allows for measurement control, data storage and processing on a connected laptop. Calibrations prior to measurement start can be launched manually or automatically. Data quality control is provided by the software, re-calibration is initiated if predefined thresholds are undershot (e.g. number of atoms).

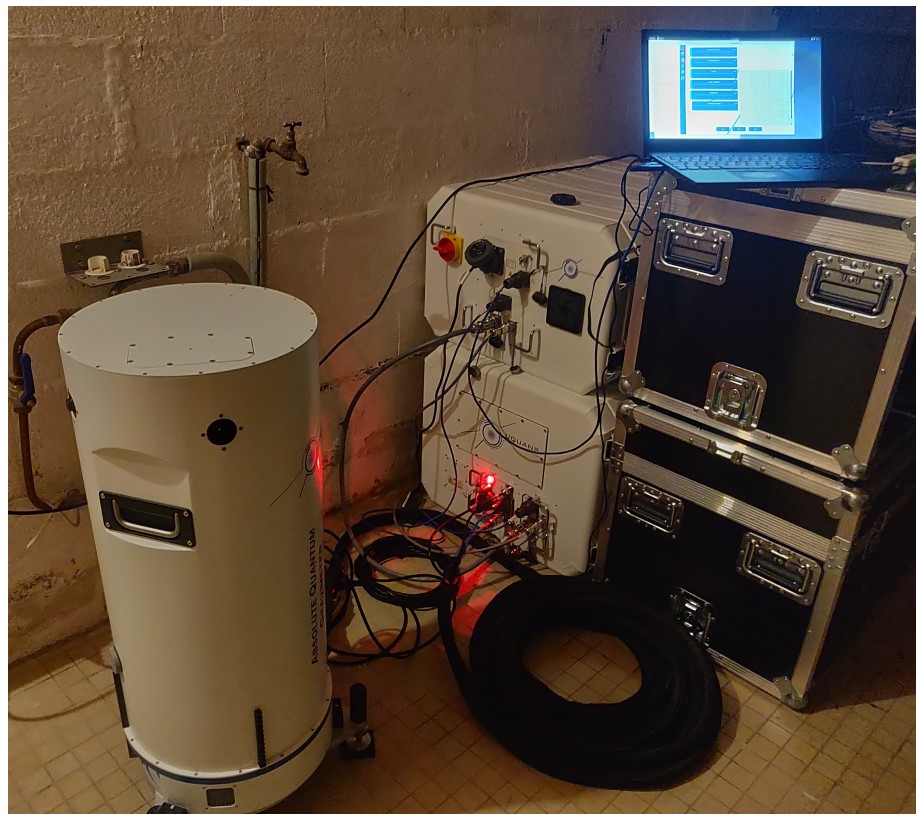

**Figure 1.** AQG-B01 in a garage in Montpellier in June 2020. The sensor head (left) is connected with the laser- and eletronic systems by 5 m of cables. The AQG GUI software is used on a laptop to control the measurement as well as to correct, visualise and analyse the gravity data.

In late April 2020, additional AQG#B01 and FG5#228 measurements and tests were continued in the facilities of the laboratory Géosciences Montpellier in the cellar, about 100 kilometers southwest of the Larzac site. The laboratory is in an urban area on the Montpellier University campus. Previous FG5#228 measurements show small gravity changes (less than $50\,\mathrm{nm\,s^{-2}}$ over one year) (Jacob et al. (2008)) for a close-by site on campus. Environmental noise is monitored with a large band seismome-
5   ter. Due to the Covid-19 lockdown, the environmental noise is largely reduced: less difference in noise level is seen between workdays, weekends, and public holidays.

On 2020-06-02 the AQG-B01 was transported from the Géosciences laboratory to a private garage in a residential area in Montpellier (in approximately 3 km distance to Géosciences).

## 3   Methods and experiments

10   The experiment timeline and data availability are displayed in Figure 2. The iGrav#002 data are available continuously. Software malfunctioning or updates and seismic events caused data gaps in the AQG#B01 series. In late January a seismic event

made a restart necessary and caused no damage to the instrument. Improvements to avoid loss of measurements caused by these incidents are in progress. An instrument test was conducted remotely by the developer on March 23rd. Apart from those, an offset of $100\,\mathrm{nm\,s^{-2}}$ was observed to emerge in the AQG#B01 gravity time series before the second temperature test on 2020-02-10. The cause is still under investigation and the authors are in contact with the instrument developer. Additional

5    monitoring variables registered during operation are being investigated. Up to this point, the main hypothesis is mechanical stress in the sensor head, acquired in-between temperature tests.

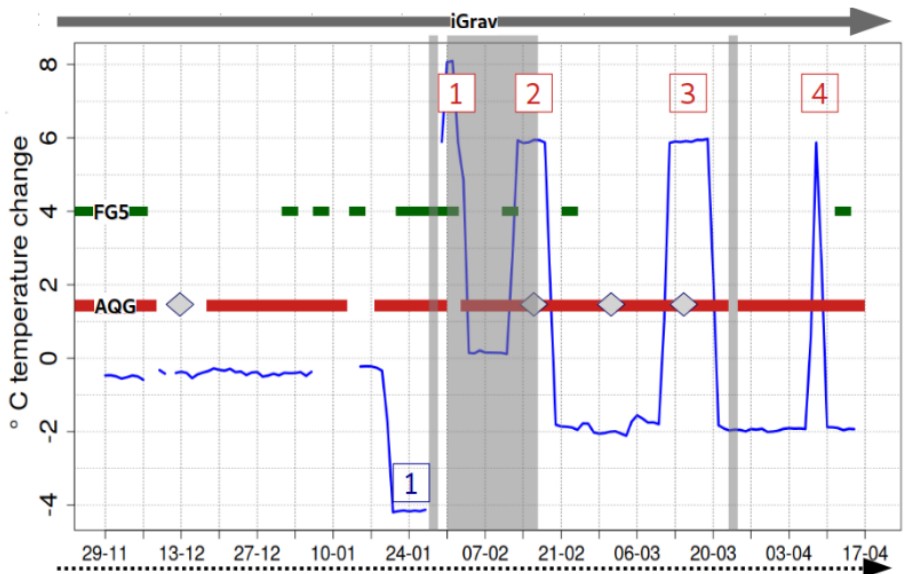

**Figure 2.** Timeline and data availability for the campaigns at Larzac observatory December 2019-April 2020. The AQG-B01 data series are marked in red, the FG5-228 measurements are marked in dark green. The iGrav-002 measured continously (marked in darkgrey). AQG-B01 data at dates marked in gray in later Janaury, early Feburary and late March were not considered in the analysis, due to inconsistencies during the experiments, as explained in section 4.4.2. The thin blue line represents the residuals from the mean observatory temperature to illustrate the temperature tests. One test has been carried out at reduced temperatures and four at increased temperatures. Grey diamonds during the AQG-B01 series mark the dates of tilt tests carried out on the AQG-B01.

## 3.1   Drift, accuracy and sensitivity

The sensitivity is assessed by calculating the Allan deviation (Allan (1966)) of the gravity residual signal. The Allan deviation is calculated for the three gravimeters (FG5#228, AQG#B01, and iGrav#002) after classical post-processing: calibration and

10    drift correction of the iGrav#002 data set, correction for solid Earth and ocean loading tides using Tsoft (Camp and Vauterin (2005)), atmospheric pressure and polar motion for all data sets. Polar coordinates were obtained from the International Earth rotation and Reference systems Service. Site-specific combined ocean and solid Earth tidal parameters had been estimated with ETERNA (Wenzel (1996)) software based on long-term iGrav#002 time series (Fores et al. (2016a)). Gravity residuals refer

hereafter to the processed gravity data set. For the AQG#B01, the mentioned corrections are carried out using the AQG GUI software.

The influence of global, non-local hydrological and atmospheric gravity effects on the Larzac site was estimated using the EOST loading service (Boy and Hinderer (2006); Boy and Lyard (2008); Boy et al. (2009)) applying the model GLDAS/Noah

(v2.1) (Rodell et al. (2004)). Gravity residuals obtained from AQG#B01 and iGrav#002 were related to local cumulative precipitation obtained from on-site rain gauges to assess the detectability of small hydrogeological mass changes. A 1D hydrological model using rainfall as input describes the gravity changes due to hydrological mass changes adequately (Fores et al. (2016a)), hence to display the gravity changes caused by rainfall an infinite homogeneous Bouguer anomaly was assumed. The Bouguer plate was calculated according to:

$$\delta g_B = 2\pi\rho G H. \tag{1}$$

G refers to the gravitational constant, $\rho$ to the density of water and the plate thickness H refers to the cumulative rainfall. The Bouguer plate equivalent was corrected for the estimated averaged daily deep percolation discharge of one mm per day (Fores et al. (2018)). The investigated precipitation event took place during the winter months, evapotranspiration was thus not considered. The FG5#228 measurements provide absolute reference points to assess any drift with time in the AQG#B01 time

series. The period between the 2020-11-28 and 2020-01-25 was used for drift assessment, as numerous tests (tilt, temperature) were conducted afterwards. Daily averaged residuals are compared in order to assess the accuracy of the AQG#B01.

The difference of effective measurement height requires a correction for the vertical gravity gradient when the accuracy of the AQG#B01 is estimated. For this set-up, the AQG#B01 effective measurement height was at 65.1 cm, the FG5#228 at 121.77 cm. The vertical gravity gradient required to correct for the vertical gravity differences between the FG5#228 and AQG#B01

measurement locations in Larzac and Montpellier have been estimated with a relative Scintrex CG5 and CG6 gravimeter. The vertical gravity gradients were estimated from measurements on two heights of 1.2 m vertical difference. Estimated vertical gravity gradients at the Montpellier site were found to be approximately -2.9 kE (1 kE = $10\,\mathrm{nm\,s^{-2}\,cm^{-1}}$). In the Larzac observatory estimated vertical gravity gradient on measurement point p1 (FG5) is -3.226 kE with a standard deviation of 0.022 kE and -3.220 kE with a standard deviation of 0.017 kE for measurement point p2 (AQG), respectively, averaged over one year

of monthly measurements (Cooke et al., in preparation). AQG#B01 and FG5#228 gravity residuals were thus transferred to the same height by correcting for a vertical gravity gradient of -3.22 and -3.225 kE in the Larzac Observatory and for -2.86 and -2.89 kE in the laboratory in Montpellier, respectively.

### 3.2 Adjustment of ambient temperature in the observatory

The observatory is kept at relatively stable 24°C in the weeks before and in between the experiments. The AQG#B01 was

operated during five periods of modified ambient temperature with periods of standard temperature in between (Figure 2). The temperature in the observatory was changed by adjusting the air conditioning device. The first temperature test comprised a reduced temperature, followed by four tests of higher temperatures, relative to the 24°C default temperature listed in Table 1.

**Table 1.** Temperature test periods in the Larzac observatory

| # | Start date | End date | Temperature °C |
|---|---|---|---|
| cooling | | | |
| 1 | 2020-01-20 | 2020-01-28 | 20 |
| heating | | | |
| 1 | 2020-01-28 | 2020-02-03 | 30 |
| 2 | 2020-02-12 | 2020-02-19 | 30 |
| 3 | 2020-03-11 | 2020-03-21 | 30 |
| 4 | 2020-04-07 | 2020-04-09 | 30 |

During the first period of increased temperature, an elevated noise level and interruption due to the seismic event were observed by both the iGrav#002 and the AQG#B01 and these periods were therefore not considered in the analysis.

### 3.3 Tilt calibration under adjusted ambient temperatures

Instrument tilt variations are monitored by a tiltmeter located at the top of the sensor head. This allows for a constant monitoring and in-line post-correction of $g$ values according to the effective tilt experienced by the instrument. Mechanical alignment between the vertical measurement axis and the tiltmeter reference is susceptible to drifts. Uncorrected tilts lead to the measurement of a projection of $g$ as compared to a well-aligned, vertical measurement. The tiltmeter reference offset can be assessed using gravity measurements and directly used for gravity correction in the AQG acquisition software. For this calibration, several measurements (~20) are performed for various x and y tilt values, up to one mrad from the initial position. The instrument's tilts in x and y are adjusted manually to reach a certain angle $\theta$ and the raw gravity data are then adjusted along a function of $\frac{1}{\cos(\theta - \theta_0)}$, in order to evaluate $\theta_0$, corresponding to the real vertical axis. The offset coefficient has been tested in the Muquans facilities in Talence (France) and has since been redone twice at the Larzac observatory. Furthermore, it was investigated whether the obtained offset coefficient had changed over time or had shown any response to temperature changes. The offset calibration test on 2020-02-19 was carried out in the GEK at an increased temperature of 28°C and 2020-03-05 at 22°C.

### 3.4 Manual tilt deregulation under adjusted ambient temperatures

Temperature changes or temperature gradients may influence mechanical parts and tilt the instrument. To investigate possible interactions between temperature and tilt and to ensure reliable application of their corrections, manual tilt deregulation was carried out during phases of modified ambient temperature. Between 2019-12-09 and 2019-12-13, manual tilt deregulation in x and y during room temperature was tested. On March 11th the room temperature was modified from 22 to 30°C and on the same day, the tilt in x-direction was manually set to 0.5 mrad.

### 3.5 Operation under semi-terrain conditions

As an intermediate step between controlled observatory conditions and open-air measurements, the influence of diurnal temperature changes and anthropogenic noise was assessed by operating the AQG#B01 in a garage in a residential area in Montpellier from 2020-06-02 to 2020-06-17. The garage is connected to a house inhabited by three people. Initially, the garage doors were kept closed. The garage is not air-conditioned. To allow for a larger amplitude of diurnal temperature change, the doors were opened in the late evening and early morning starting from 2020-06-08.

During the last days of operation, several acquisition stops occurred that required a manual measurement relaunch and causing data gaps mainly during the night. The experience obtained from the operation in the garage shows that frequent measurement stops required software improvements and modifications in the response to instrumental variables in order to create a robust, continuous measurement system without the necessity of regular human intervention. These software updates have meanwhile been implemented.

### 3.6 Repeatability

On 2020-04-17 the AQG#B01 was transported to the facilities of the laboratory Géosciences Montpellier on University campus and operated in the basement of the building. The distance between the Larzac observatory and Montpellier is about 100 km and there is 640 m difference in altitude. The transport to Montpellier was the second displacement of the AQG#B01 after its first delivery to the Larzac observatory in November 2019. This implies the turn-off, deconnection, displacement and cold restart of the instrument. The data were compared to FG5#228 measurements at both sites. Small-scale repeatability was assessed using repeated gravity measurements on the same position in the gravity lab in the basement of Géosciences Montpellier. In between these measurements, the instrument was not turned off or disassembled, only the measurement was temporally stopped and the sensor head was moved within the room. Smale-scale repeatability was therefore assessed under warm conditions without a full restart of the system. Vertical gravity gradients were additionally estimated with a relative gravimeter (Scintrex CG6).

### 3.7 Coriolis effect

Gravimeters are sensitive to a Coriolis shift, the Sagnac effect caused by the Earth's rotation. This effect can generate an additional bias in quantum interferometers. The horizontal atomic velocity component generates an additional Coriolis acceleration depending on the E-W direction. This leads to a possible gravity bias (Peters et al. (2001), Louchet-Chauvet et al. (2011)). By symmetrical construction (hollow pyramidal reflector and location of the detection photo-diodes), horizontal atomic velocities are reduced and the AQG#B01 should only show a negligible sensitivity to the Coriolis effect as in the CAG gravimeter (Louchet-Chauvet et al. (2011)). We assess the potential residual Coriolis effect in the AQG#B01. By rotating the device by 180 $^\circ$, two opposite orientations are obtained, hence a change in sign of the Coriolis acceleration is expected. As in Louchet-Chauvet et al. (2011), the tests were carried out under the assumption that parameters do not change between the measurements. The same set-up was thus kept constant to rule out other systematic effects. Coriolis AQG#B01 measurements last at least 24 hours to reduce the effect of residuals after tidal correction.

## 4 Results and discussion

### 4.1 Sensitivity

The sensitivity of the AQG#B01 is firstly evaluated by statistical time series analysis in comparison with other gravimeters and secondly by direct monitoring of natural gravity changes. It was calculated for the first week of December 2019 using 10 minute AQG#B01 data and one-minute iGrav#002 data (Figure 3). The first week in December 2019 shows no rainfall and no instrument tests were performed. At an integration interval of one hour, the sensitivity of the AQG#B01 reached $10 \, \mathrm{nm \, s^{-2}}$, the iGrav#002 shows a higher sensitivity at short time scale, but both the iGrav#002 and the AQG#B01 reach the same level of sensitivity at 24 hours. All three instruments show a slight increase at long time likely due to environmental noise and tides residuals.

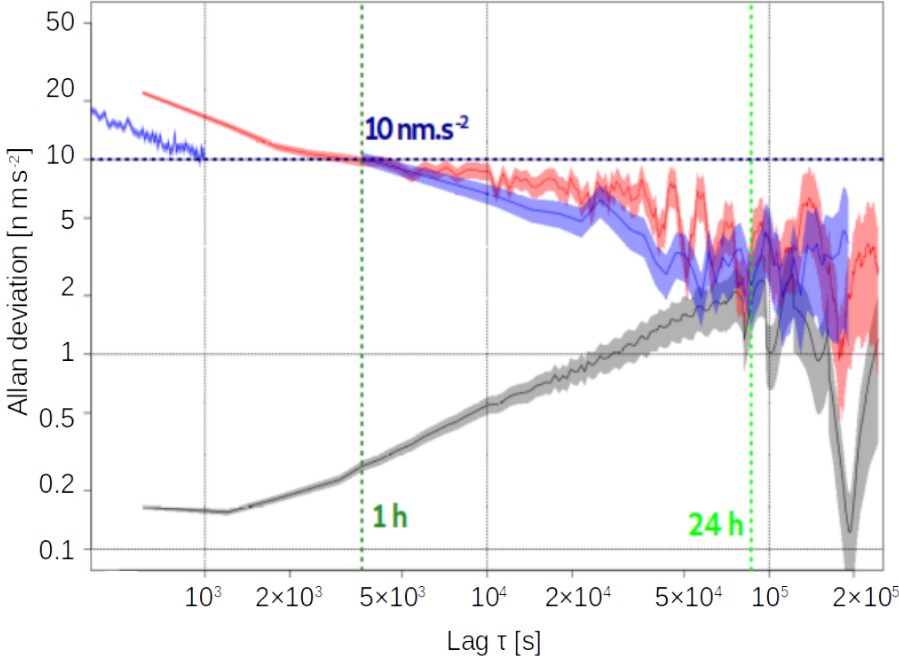

**Figure 3.** Allan deviation in nm s$^{-2}$ for the first week of December 2019 at the Larzac site. One-minute series were used for the iGrav-002 (grey), 10-minute data for the AQB-01 (red). The irregular sampling frequency of the FG5-228 (100 drops every 10 seconds, then 2600 s break) was taken into account by plotting the short-term Allan deviation (< 1000s for 10 s data) and the long-term (1 h data) separately (both blue). The horizontal blue dashed line shows the sensitivity benchmark of 10 nm s$^{-2}$, the dark green vertical dashed line signifies the integration period of 1 h, the light green one that of 24 h.

The analysis of the Allan deviation showed that for averaging periods of a few hours the iGrav#002 is the more sensitive instrument for this data set. The AQG#B01 and the iGrav#002 converge to show similar sensitivity of clearly better than $10 \, \mathrm{nm \, s^{-2}}$ at a 24 h averaging period. For shorter integration intervals the sensitivities of the FG5#228 and AQG#B01 are

comparable and lower than that of the iGrav. The AQG#B01 does not achieve the sensitivity of laboratory quantum gravimeters that have achieved $2\,\mathrm{nm\,s^{-2}}$ in less than 2000 s (CAG; Gillot et al. (2014)) or a mobile quantum gravimeter for which $0.5\,\mathrm{nm\,s^{-2}}$ after one day have been reported (GAIN; Freier et al. (2016)).

For measurements longer than one day, the AQG#B01 is likely to be equally sensitive as the iGrav#002. To obtain values closer to the possible highest sensitivity, a prolonged measurement of several weeks during a low noise period of stable weather conditions and little human interventions is required, as it would be possible for example during summer months in the Larzac observatory. The Allan deviation of the AQG#B01 data recorded in Montpellier showed a sensitivity of approximately $20\,\mathrm{nm\,s^{-2}}$ after one h, after 24h it was below $10\,\mathrm{nm\,s^{-2}}$. This decrease in sensitivity for the urban site compared to the Larzac observatory can be explained by the higher level of environmental noise in the university building.

A sensitivity of $10\,\mathrm{nm\,s^{-2}}$ is achieved in one hour in a naturally low noise environment and a sensitivity of $20\,\mathrm{nm\,s^{-2}}$ in an urban environment. In the context of characterising the AQG#B's sensitivity, the use of rubber pads below the tripod feet to reduce the effect of ground vibrations was assessed for future field experiments. Figure 4 shows that the Allan deviation is reduced for measurements of less than an hour. At one hour duration, there is no significant difference, for longer integration times there is likely no major improvement. This needs to be reassessed for longer series and at periods of higher environmental noise as the activity at the university was reduced during the COVID19-lockdown.

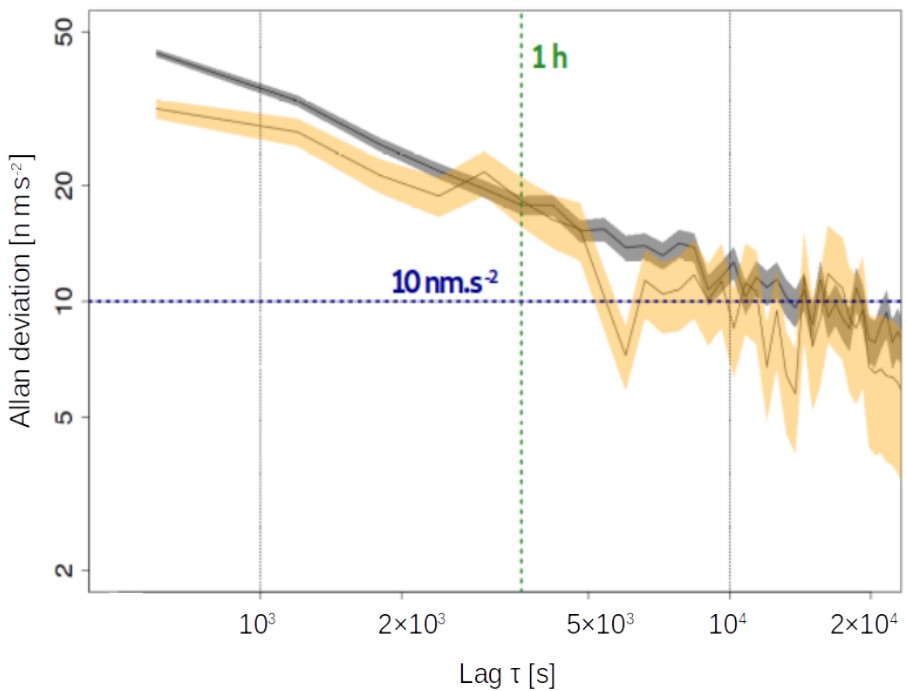

**Figure 4.** Comparison of sensitivity with (orange) and without (grey) placing the AQG-B01 tripod on rubber pads. The measurement without rubber pads were conducted on 2020-04-23 - 26, without pads on 2020-04-28.

## 4.2 Sensitivity to hydrogeological gravity changes

A significant increase in $g$ of $\sim 80\,\mathrm{nm\,s^{-2}}$ between the 2020-12-13 and the 2020-12-27 has been detected by the three gravimeters. As can be seen on Figure 5 (a) daily gravity residuals of all of them show high resemblance in their temporal variations and differ within their error margins.

Figure 5 (b) shows the series of rainfall events in 2019. All three gravimeter signals follow the increase in gravity caused by the rise in soil water content in the aftermath of the rainfall events. The gravity time series continues to increase even after rainfall stops. This is expected since the infiltrating water moves further into the gravimeter's spatial sensitivity which can be described as two flat cones above and below with the instrument in the centre.

The measured change in $g$ is comparable to the expected increase caused by the corresponding Bouguer plate equivalent of the rainfall event. Figure 5 (c) shows the Bouguer plate equivalent in $\mathrm{nm\,s^{-2}}$ adjusted for deep discharge. Differences between the gravimeters' responses to the rainfall event can partly be explained by their position within the observatory building and the heterogeneity of rock properties. Local gravity measurements are impacted by the building due to shielding from precipitation, commonly referred to as the "umbrella effect" (e.g. Creutzfeldt et al. (2008); Deville et al. (2012); Hector et al. (2014); Fores et al. (2016a); Reich et al. (2019)). The AQG#B01 is operated on measurement point p2 in one corner of the building and hence more exposed to the area outside the building. The umbrella effect admittance can be calculated for sensor height and the specific measurement location within the building (Deville et al. (2012); Fores et al. (2016a)).

An umbrella effect of the Larzac observatory e.g. of 80% difference in the gravity of a given Bouguer plate measured on the central pillar (iGrav) inside the building has been observed by Fores et al. (2016a) and decreases once the infiltration front moves further in-depth. For the given Bouguer plate equivalent of $80\,\mathrm{nm\,s^{-2}}$, up to $64\,\mathrm{nm\,s^{-2}}$ difference in $g$ can be expected for the iGrav. For the AQG#B01 pillar an umbrella admittance of 46 % has been estimated from a truncated Bouguer plate. Umbrella effect related differences between AQG#B01 and iGrav#002 could hence be up to several tens of $\mathrm{nm\,s^{-2}}$. As the umbrella effect depends on the initial conditions and the previous rainfall events, it is difficult to determine the sign of the relative offset between the AQG#B01, FG5#228 and iGrav#002 series without further information.

To summarise, the AQG#B01 clearly measures the gravity increase of less than $100\,\mathrm{nm\,s^{-2}}$ after the rainfall event in the same range as the iGrav. The gravity changes are coherent with previous studies at the site. The differences between the iGrav#002 and AQG#B01 data can be explained by limits of the sensitivity of the AQG#B01 and the heterogeneity of the hydrogeological context in a karst area.

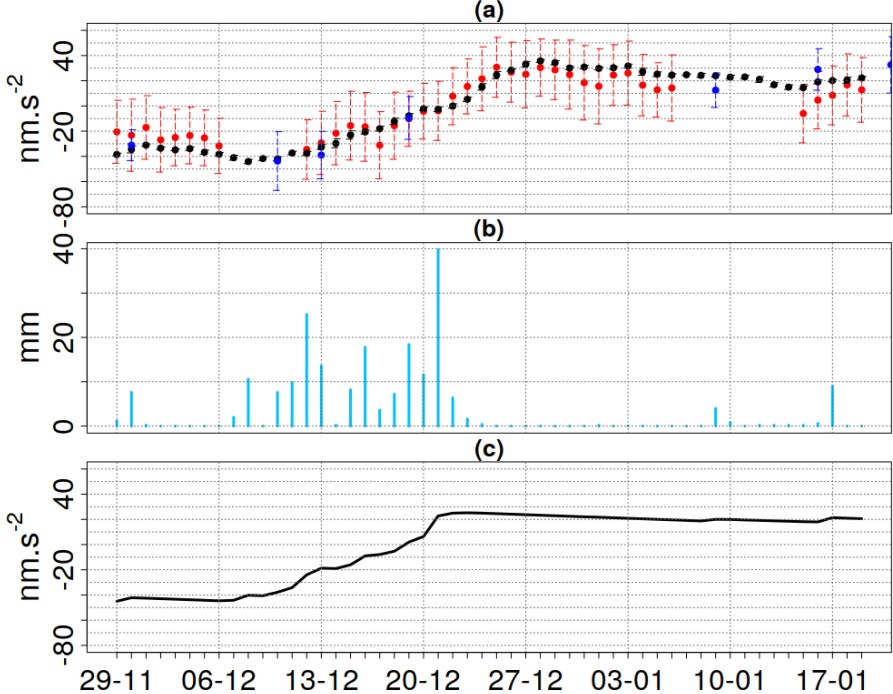

**Figure 5.** (a): AQG-B01 (red), FG5-228 (blue) and iGrav-002 (black) daily gravity residuals at the Larzac observatory in nm s$^{-2}$. Residuals of $g$ refer to the mean of the shown measurements, respectively. The error bars refer to the standard deviation of 24 h of measurement. (b): Rainfall series 2019-12 – 2020-01 on the GEK, LaJasse site, L'hospitalet-du-Larzac. (c): Bouguer plate equivalent (in nm s$^{-2}$) of precipitation corrected for deep discharge.

### 4.3 Accuracy, repeatability, and drift

The differences between the AQG#B01 and the FG5#228 during 2 months yielded an estimated, statistically insignificant drift of -0.02 $\pm$ 0.04 nm s$^{-2}$ per day. A longer measurement is currently in progress to investigate a potential long-term drift. Establishing the complete accuracy budget is a complex task for a new instrument and remains work in progress. A first

5 approximate estimation of the accuracy is done by comparing the AQG measurements with the ballistic gravimeter (FG5#228) in the Larzac observatory. Daily AQG#B01 and FG5#228 gravity residuals show high resemblance in their temporal variations and differ within their error margins (Figure 5). The difference between AQG#B01 and FG5#228 based on 13 measurements (daily averages) between December 2019 and April 2020 in the Larzac observatory is on average 110 nm s$^{-2}$ with a standard deviation of 31 nm s$^{-2}$, with the FG5#228 values being smaller than those measured with the AQG#B01. For the Montpellier

10 laboratory, the difference between both instruments is based on 24h averages of ten AQG#B01 measurements between 2020-04-27 and 2020-05-14 and one FG5#228 measurement on 2020-06-10. The difference showed 44 nm s$^{-2}$ with a standard deviation of 66 nm s$^{-2}$, with the FG5#228 values being higher than those of the AQG#B01.

**Table 2.** AQG-B01: Small-scale repeatability in Montpellier in 2020. Differences as residuals from the mean of the listed measurements, based on 24h averages. The standard deviation refers to 24 h.

| Date | $\Delta g$ [nm s$^{-2}$] | SD [nm s$^{-2}$] |
|------|------|------|
| 2020-04-27 | -43 | 46 |
| 2020-04-28 | -5 | 36 |
| 2020-05-03 | 19 | 41 |
| 2020-05-08 | -8 | 110 |
| 2020-05-13 | 27 | 66 |
| 2020-05-27 | -9 | 60 |

Absolute comparison between both instruments is limited due to the set-up on different locations on the pillars and is impacted by the uncertainty related to the vertical gravity gradient (VGG) correction. An offset in vertical gravity gradient correction between FG5#228 and AQG#B01 of 10 E for the difference in height between the two instruments' sensors ($\delta_{height}$ = 56.67 cm) accounts for an uncertainty of $5.7\,\text{nm}\,\text{s}^{-2}$. The VGG for the pillars in the observatory can be estimated by repeated

relative gravimeter measurements on different heights and their uncertainty has been estimated to be around 20 E (Cooke et al., in preparation). Hence, about $11\,\text{nm}\,\text{s}^{-2}$ of uncertainty is due to the fact that the VGG cannot be estimated more precisely up to this point. The possibility of a more precise estimation of the VGG employing AQG measurements on tripods is being discussed and in preparation. Likely due to soil moisture changes, VGG at the Larzac site may vary by several tens of E over weeks and months (Cooke et al., in preparation). We recommend to take the accuracy assessment at the Larzac site with caution

and to repeat the measurements together with the FG5 in the near future.

Small-scale repeatability tests were only carried out in Montpellier. Table 2 shows an average small-scale repeatability of $3\,\text{nm}\,\text{s}^{-2}$ with a standard deviation of $25\,\text{nm}\,\text{s}^{-2}$ for repeated measurements on the same point and orientation after returning from displacements and other experiments.

AQG#B01 was operated on two measurement points within the same room in the Géosciences laboratory at about one meter

distance, of which one serves FG5#228 measurements. At this short distance, no considerable horizontal difference in $g$ is expected. CG6 relative gravity measurements were carried out with two instruments (CG6#120 and CG6#125) on 2020-03-26 on both points and found a negligible difference in $g$ of $2 \pm 6\,\text{nm}\,\text{s}^{-2}$. The difference in $g$ measured with the AQG#B01 between point 1 and point 2 is $15 \pm 48\,\text{nm}\,\text{s}^{-2}$. The measurement on point 2 was carried out using rubber pads under the tripod. The rubber pads has not significant effect in the 10 min averaged gravity readings. Only a small decrease of the gravity

readings dispersion at short time scale (< 10 minutes) was observed.

To summarise, these first results show a repeatability of $3\,\text{nm}\,\text{s}^{-2}$ with a standard deviation of $25\,\text{nm}\,\text{s}^{-2}$ and no detectable drift over 2 months operation. No impact of transport and displacement such as mechanical relaxation on the gravimeter was observed for small (one m) or large (100 km by car) distances. The complete accuracy budget is still under investigation. Additional location changes between the sites with both the AQG#B01 and the FG5#228 are required to reliably quantify the

repeatability. Due to the potential error margin caused by the earlier mentioned uncertainty introduced by the differences in vertical gravity gradient correction and measurements on different pillars, the accuracy in relation to the FG5#228 cannot be estimated completely up to this point.

## 4.4 Temperature and tilt

### 4.4.1 Tiltmeter calibration

The stability of the tiltmeter offset is important both for long term monitoring and for the repeatability of the measurements after transport by car. Tilts of 5 µrad lie probably within measurement uncertainty. A tilt bias of 50 µrad leads to an error of $12.3\,\mathrm{nm\,s^{-2}}$ and at small angles increases in a quadratic way which leads to $49\,\mathrm{nm\,s^{-2}}$ for 100 µrad. The tiltmeter offset calibration changed by 4.7 µrad for x and by 3.8 µrad for y between Talence and the Larzac observatory ($\sim 400$ km) which is within measurement uncertainty. The tilt calibration obtained from data acquired in the GEK at 30°C on 2020-02-19 yields very similar results to the calibration carried out on 2019-10-31 in Talence at 20°C. On 2020-03-05 a partial test of certain chosen tilts at 22°C was carried out that showed coherent results to the test carried out at 30°C on 2020-02-19. The tiltmeter offset calibration carried out in Talence and in Larzac lie almost four months apart and include the transport of the instrument from Talence to the Larzac site. The obtained offset values showed hence very minor changes over time, after transport or at different external temperatures. The data showed that the tilt calibration is likely to be independent of temperature and to stay stable over time.

### 4.4.2 Influence of temperature

During the temperature experiment AQG#B01 residuals did not show any statistically significant correlation with external temperature. Gravity residuals, external temperature and tilts are displayed in Figure 6. The AQG#B01 showed no significant shift in the range of 20 to 30°C nor during episodes of tilt change. The visible variations lie within the earlier observed variations and uncertainty range. Relative offsets between AQG#B01 and iGrav#002 can be linked to the umbrella effect and karst heterogeneity as discussed in section 4.2. Precipitation was negligible during these measurements and thus no significant effect of precipitation was observed in the iGrav#002 monitoring. Temperature's impact on the AQG#B01 gravity residuals was difficult to assess considering the first two episodes of elevated temperatures. Experimental conditions varied between the tests, the results were not conclusive. Possibly due to the malfunctioning of the air conditioning, three spikes in temperature occurred during the first test. On 2020-01-31, 2020-02-01, and on 2020-02-03 approximately 35°C were reached for about eight hours each time, before decreasing again to settle at 30°C. During the second temperature test an insulating cover around the sensor head had not been removed resulting in an increase of temperature in the sensor head above nominal operation conditions. The sensor head can be easily wrapped in the fitting insulation (down) cover.

AQG#B01 and iGrav#002 residuals differ less than $50\,\mathrm{nm\,s^{-2}}$. The reduced temperature test (20°C; January 20-28) did not cause any significant gravity response to increased external temperatures. In March and April, the tilts stay stable before,

during, and after the temperature test, and no remarkable shift between the residuals of $g$ obtained with the AQG#B01 compared to those obtained with the iGrav#002 is observed (Figure 6).

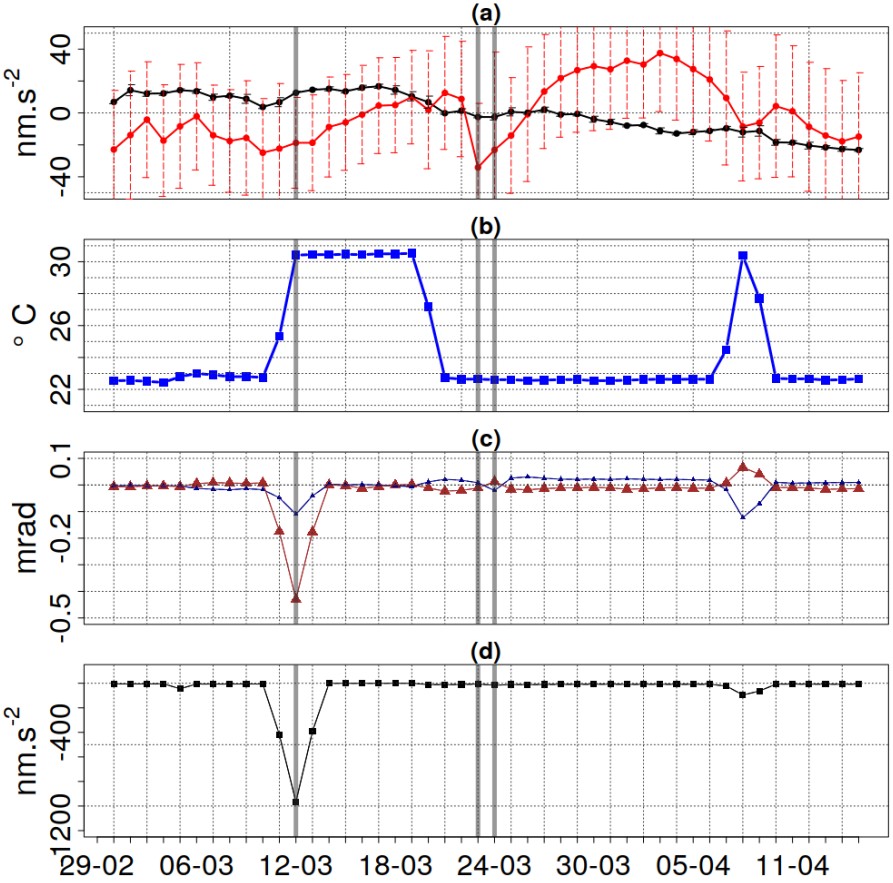

**Figure 6.** (a): Residuals of $g$ for the iGrav-002 (black) and AQG-B01(red). (b): external temperature (blue). (c): Tilts in x (brown) and y (dark blue) for the tests 3 and 4 in March and April 2020 (see Figure 1 for the timeline). (d): Tilt correction combined for x and y in nm s$^{-2}$. Dates marked as grey were subject to further instrument tests and adjustments.

### 4.4.3 Combined tilt and temperature tests

Manually adjusted tilts during temperature changes did not show a visible change in $g$ recorded by the AQG#B01. As can be seen in Figure 6, the tilts showed a minor response during the last temperature test in early April. No impact on the corrected $g$ value (with a tilt correction of $\sim 1000$ nm s$^{-2}$) is observed. Tilts return to values close to zero after a change in temperature.

Simultaneous manual tilt deregulation and room temperature change did not lead to any clear shift of the difference in $g$ between the value of the AQG and that of the iGrav. This result suggests that the measurement of $g$ is not impacted by temperature. It cannot be ruled out that tilts of more than one mrad require a different correction than small tilts. It is thus recommended

to keep the sensor head well levelled during operation. The gravity series obtained with the AQG#B01 before, during, and after an elevated temperature of 30°C in March and April 2020 show no impact of these. To reproduce these findings, further temperature experiments and larger ranges should be carried out, potentially exploring also much lower temperatures.

### 4.4.4 Operation under semi-terrain condition

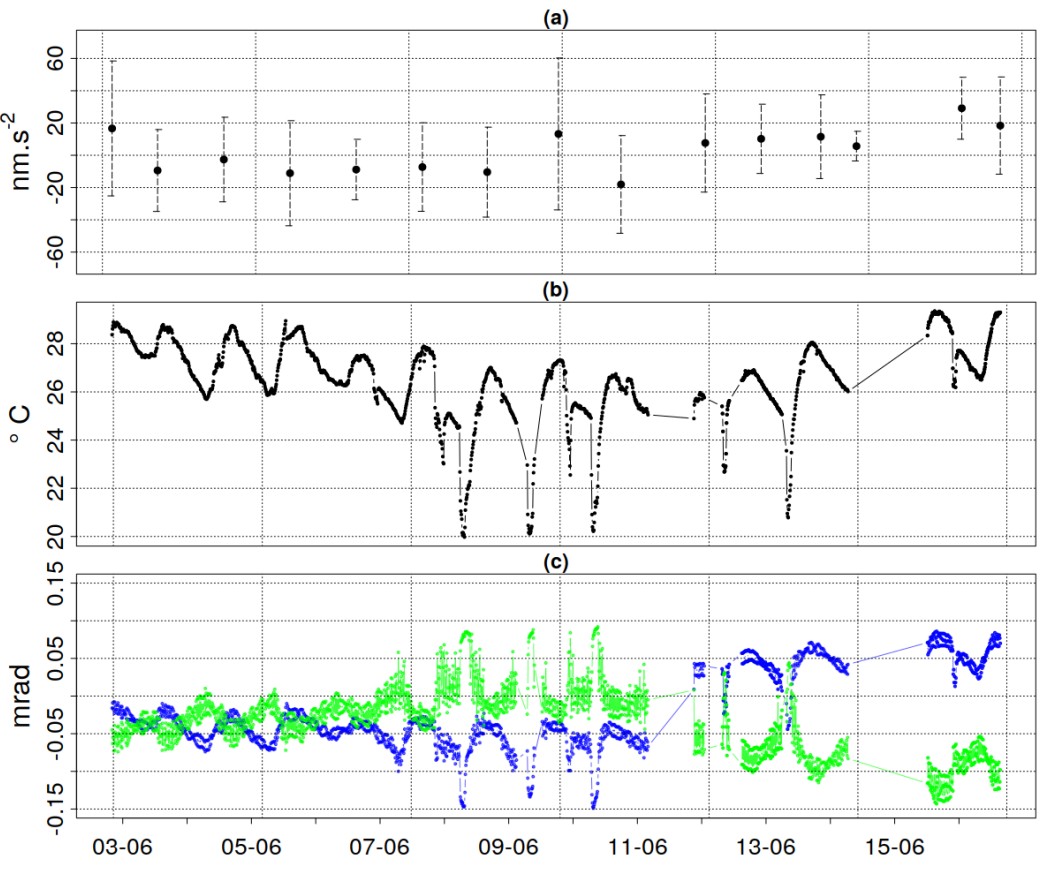

**Figure 7.** (a): AQG-B gravity residuals from the mean recorded during operation in a garage in Montpellier 2020-06-02 – 2020-06-17. (b): external temperature in the garage recorded by AQG-B01 in °C.(c): Sensor head tilts in x (blue) and y (green) given in mrad.

5  Daily averages of $g$ show a standard deviation of less than $50\,\mathrm{nm\,s^{-2}}$, differences between daily averages during the two weeks of operation lie within statistical uncertainty (Figure 7, (a)). Temperatures in the garage vary between 20 and 30 °C. Strong and fast temperature drops were a result of opened garage doors. As can be seen in Figure 7 (c), during periods of external temperature a change in sensor head tilts occurs. Tilts returned to values close to zero within hours. Maximal inclinations of about 0.15 mrad were found for tilts in x-direction. Tilts in y-direction stayed more stable and show generally fewer variations.

10  After a manual re-calibration of the tilt after 2020-06-11, tilts move in opposite directions than before, which is likely related

to a change in equilibrium after each re-calibration. The residents entered the garage several times throughout the day from the interior door. The opening of the outward garage doors for a temperature change caused increased air circulation in the room. Possible impacts of these winds on tripod levelling and imbalance in temperature of different sides of the tripod cannot be ruled out and need to be investigated. We did not explicitly measure wind or humidity changes. However, opening the doors
5   clearly caused noticeable air circulation in the room. During those two weeks, both dry and sunny as well as very rainy weather conditions have been observed without significant variations in the measured gravity by the AGQ.

As expected, the sensitivity(Figure 8) is reduced compared to the Larzac site. $26\,\mathrm{nm\,s^{-2}}$ are reached after one hour of averaging, sensitivity is better than $10\,\mathrm{nm\,s^{-2}}$ after 24 h of averaging.

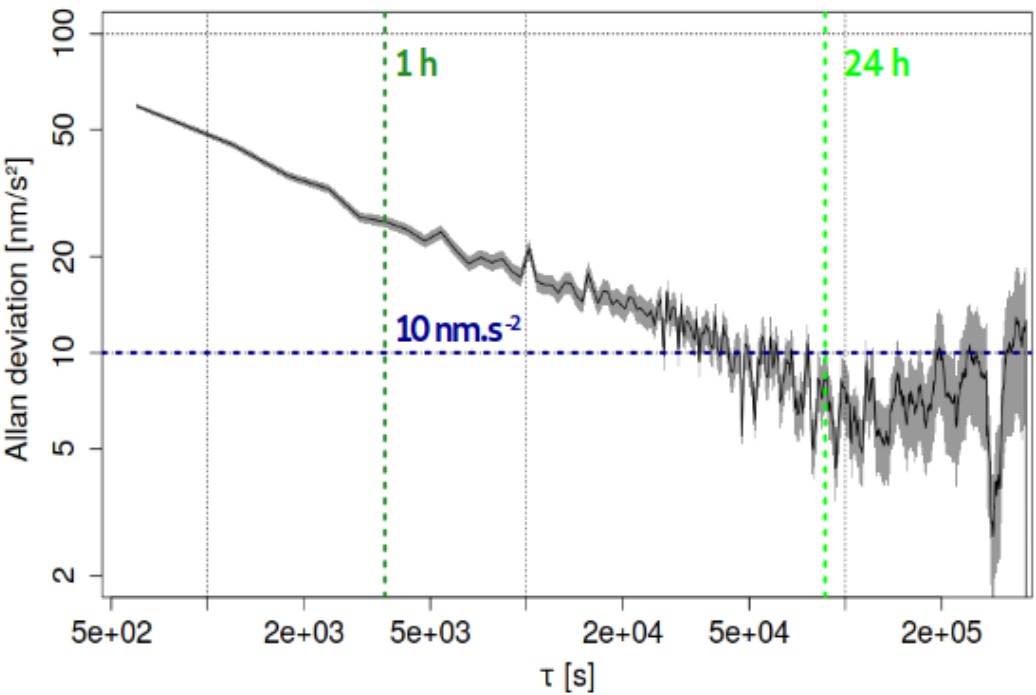

**Figure 8.** Allan deviation of 10 min AQG-B01 data obtained during operation in a garage in Montpellier. The horizontal blue dashed line shows the sensitivity benchmark of $10\,\mathrm{nm\,s^{-2}}$, the dark green vertical dashed line signifies the integration period of 1 h, the light green one that of 24 h.

These results demonstrate the successful deployment of the AQG#B01 in an urban environment. Stable measurements within
10   $\pm\,20\,\mathrm{nm\,s^{-2}}$ were achieved during two weeks of operation in elevated noise levels and in presence of several environmental effects, in particular, temperature changes of several °C within minutes and air movement when opening the garage doors.

## 4.5 Coriolis effect

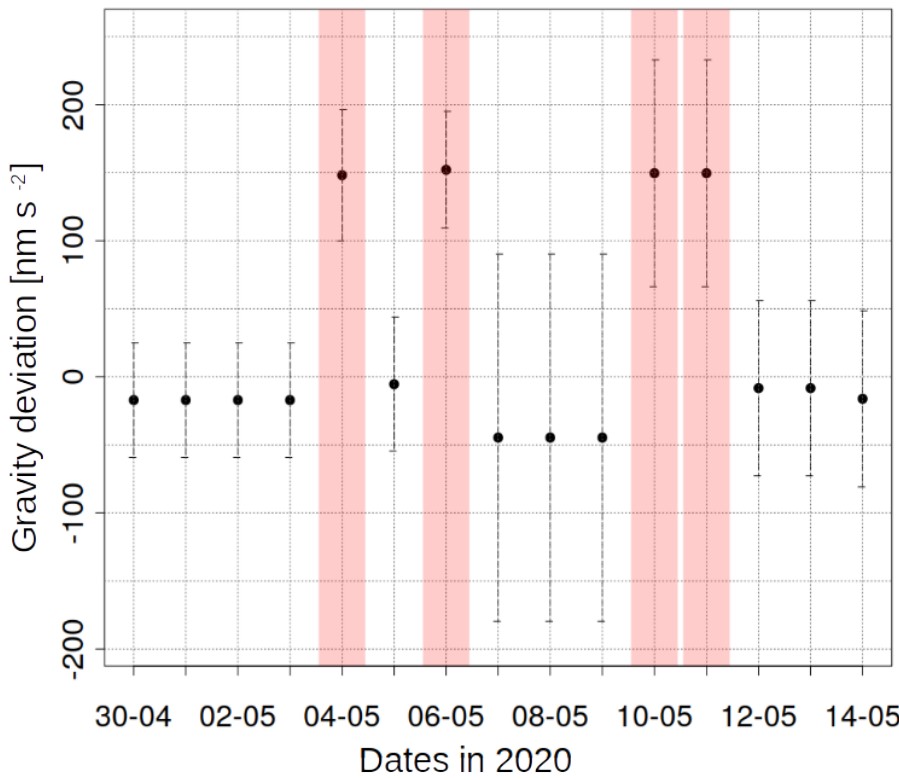

**Figure 9.** Residual gravity values in nm s$^{-2}$ obtained with the AQG-B01 during the Coriolis tests at Géosciences Montpellier in 2020: The red bars show a sensor head orientation of 180° compared to the default set-up. Error bars refer to the standard deviation during 24 h. The value of $g$ averaged over 24 h obtained on 2020-06-10 with the FG5-228 was subtracted. Horizontal grid lines show steps of 50 nm s$^{-2}$.

To assess the potential impact of the Coriolis effect caused by changes in the sensor head's orientation we carried out measurements rotated by 180° with respect to the initial position. As can be seen in Figure 9, the averaged gravity residuals for the two orientations show a $\delta g$ of approximately 150 nm s$^{-2}$ relative to the FG5#228 measurement. These results are much higher than expected, as it has not been observed in other AQG devices (Muquans, 2020, personal communication). These values are higher than the estimation of the Coriolis effect for the CAG (LNE-SYRTE), which yielded 4 nm s$^{-2}$ uncertainty with peaks of up to 60 nm s$^{-2}$ as a combination of several uncorrected effects (Louchet-Chauvet et al. (2011)).

The sign of the Coriolis offset on the AQG#B01 as compared to the absolute gravity measurement by the FG5#228 has an important implication for the accuracy assessment and the interpretation of the differences between the two instruments: The FG5 measurement lies between the AQG#B01 values for the two orientations during the tests in Montpellier. FG5#228 measurements were higher than AQG#B01 measurements in the Larzac observatory. The average difference between both instruments hence likely requires to be increased by $\sim$ 75 nm s$^{-2}$, depending on the orientation of the AQG#B01 sensor head.

Repeated and additional orientations (90° and 270°) are work in progress, as well as in comparison with other gravimeters. The authors are in contact with the developer for in-depth instrument tests. Up to this date, the exact source of this change in $g$ has not been identified yet. It is thus recommended to pay attention to sensor head orientation during operation. It will be necessary in the future to evaluate through repeated tests if the impact of Coriolis is stable over time and and whether a Coriolis
correction can be established according to the orientation. The Coriolis effect needs to be included in the complete uncertainty budget.

## 5   Conclusion and perspectives

In this study, we show the results of instrumental tests aimed at the characterization of the AQG#B01 for field applications. The AQG#B01 prevailed as a reliable instrument in controlled laboratory conditions. Over two months no significant drift was
observed and temporal variations are in coherence with the MGL-FG5. Its sensitivity after 24 h of data integration is close to that of the iGrav. In low noise environments, the AQG#B01 showed a sensitivity of $10\,\mathrm{nm\,s^{-2}}$ after 1 h. On multi-year time scales the performance of the AQG in comparison with superconducting gravimeters still requires evidence with respect to noise and resolution of sub-daily to weekly phenomena (Scherneck et al. (2020)). AQG#B01 $g$ residuals showed no correlation with manually increased tilts nor increased temperatures. An offset compared to the other gravimeters occurred and its causes
are under investigation, as is the accuracy of the AQG#B01. The AQG#B01 time series obtained in the garage during two weeks of measurement was stable and did not show any significant correlation with temperature changes.

Furthermore, the obtained results suggest its suitability for field studies, upon further testing and validation. It is suitable for operation at least within a temperature range between 20 and 30°C over several days and weeks. Tilt correction is likely to be applied correctly even for relatively large tilts and during periods of higher temperatures.
For a thorough field evaluation, it is crucial to test larger temperature ranges. Operation in a garage where air circulation and humidity changes have not been controlled suggest that the measurement is not influenced by these factors. In future experiments, the potential influence of strong winds needs to be assessed. Due to the very recent delivery of the instrument, displacements between observatory locations and system mounting and un-mounting cycles have not been as frequent and need to be repeated. Further practical aspects such as the potential influence of cable insulation, unstable power supply or tilts of the
electronic and laser systems are to be assessed in due time.

The AQG#B01 detected gravity changes caused by hydrology ranged at the same order of magnitude as the iGrav. The resemblance between AQG#B01 and iGrav#002 residuals concerning their response to a rainfall event demonstrates the AQG#B's capability to detect small transient mass changes. This speaks for the applications of the AQG#B01 in hydrogeophysical studies among others. Mapping and quantification of water storage dynamics as an example for other transient mass changes could
be one of many promising applications in this field. Jacob et al. (2010) mapped subsurface water storage heterogeneously distributed in the 100 km² karst catchment in the Larzac using time-lapse micro-gravity. Gravity changes caused by local aquifer re- and discharge dynamics reached up to $220\,\mathrm{nm\,s^{-2}}$ with a survey precision between 24 and $50\,\mathrm{nm\,s^{-2}}$. Due to the strong and non-linear drift of relative gravimeters, the analysis of spatial surveys requires a least-squares network adjustment in which

spatial and temporal gravity ties between stations within a survey network are considered (Jacob et al. (2010); Hector and Hinderer (2016); Kennedy and Ferré (2015)). The 40 gravity stations were repeatedly visited with a relative spring gravimeter which required absolute FG5#228 measurements as a reference for drift correction. Each of the four surveys yielded over 100 gravity ties as they covered all 40 stations in 12 loops consisting of five to ten stations in each loop. Their study meant seven

full days of work for two operators for each survey due to numerous returns to the reference station. Meanwhile, Fores et al. (2016b) showed that relative gravimetry with a CG5 is highly sensitive to temperature. Correcting for temperature induced effects shed new light on apparent spatial gravity differences measured in the field. It is possible that the spatial water storage heterogeneity suggested by Jacob et al. (2010) needs to be re-interpreted in view of remaining, uncorrected effects.

The applications in the mentioned studies show the potential gain in precision and time saved provided by the AQG. The

AQG#B01 allows to combine two instruments in one. In absence of a detectable drift, regular calibration is not required. In principle, no repeated loops would be necessary, as no gravity ties need to be established. The need for another indoor reference gravimeter becomes obsolete. High precision gravity acquisition is possible with this new movable instrument. It is easy to set-up and use without the need for operation and maintenance by an expert, as for the FG5. The survey time investment and data treatment could be hence reduced remarkably for spatial gravity mapping. The results so far suggest a sensitivity between

10 and 20 nm s$^{-2}$ after one hour. These first results are promising that the AQG#B01 could reach significantly higher precision than relative gravimeters while being transportable. Even if the sensitivity of the AQG#B01 during outdoor operation still needs to be investigated, the results suggest reliable operation in different temperatures, very likely reaching a higher sensitivity than that of relative gravimeters after only one hour of measurement. The repeatability has been quantified as better than 50 nm s$^{-2}$. Our study also revealed important precautions that need to be taken. The first results on the Coriolis effect suggest that for

repeated studies the same orientation of the sensor head needs to be kept.

Its time efficient deployment offers new possibilities for natural hazard monitoring and potential early warning systems, some of which are already under investigation with the AQG#A. Joint absolute and relative gravimetry monitoring of volcanic activity are studied at Mt Etna (Carbone et al. (2020)). Another recent project focuses on the AQG on a mobile facility for a hydrological extreme event task force (Reich and Güntner (2020)). The observatory tests under controlled conditions aimed

at singling out the effects of ambient conditions, mainly temperature. The next step is clearly to carry out tests outside the building. The estimation of the vertical gravity gradients by operating the AQG#B01 on two different heights would add another application to the instrument's repertoire.

*Data availability.*  The data are available at http://doi.org/10.5281/zenodo.4279110

*Author contributions.*  Nicolas Le Moigne carried out the instrumental tests. Data analysis was carried out by Anne-Karin Cooke. Planning

of experiments and interpretation of results was a joint effort of C.C., N.L.M and A.-K.C.

*Competing interests.* The study presented in this paper has been carried out in the context of the PhD thesis of Anne-Karin Cooke within the network ITN Enigma (https://enigma-itn.eu/esr-n8-monitoring-water-storage-changes-new-portable-absolute-quantum-gravimeter/). The University of Montpellier (Géosciences) and the company Muquans, the developer of the new absolute quantum gravimeter, have a collaboration for this purpose. During a large part of her PhD, she has been an employee of Muquans. Until October 2020 she was employed by the University of Montpellier.

*Disclaimer.* All measurement presented here are from the Larzac observatory hosted by OSU OREME (http://www.oreme.org) and SNO H+ (http://hplus.ore.fr/). Larzac observatory and instrumentation are mainly funded by the CNRS INSU, ANR, Montpellier University, OZCAR, RESIF and Occitanie region. ENIGMA ITN has received funding from European Union's Horizon 2020 research and innovation programme under the Marie Sklodowska-Curie Grant Agreement N°722028.

*Acknowledgements.* We would like to thank Pierre Vermeulen and Laura Antoni-Micollier from Muquans for their active support, discussion and feedback. Furthermore, we thank Sébastien Merlet for the discussion and advice. Last but not least, we acknowledge the inhabitants who allowed the experiments in the garage.

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
