# Peer review of "Evaluation of the capacities of a field absolute quantum gravimeter (AQG#B01)"

_Geoscientific Instrumentation, Methods and Data Systems, 2020_

## Referee Comment (RC1) · Anonymous Referee #1 · 1 Oct 2020

This study reports on first performance tests of a new generation of absolute quantum gravimeters that have explicitly been designed for mobile operation in field measurement campaigns. The study is of value in reporting for the first time on the measurement performance of one instrument of this type of the Muquans AQG B series, and on the possible impacts of changing operation parameters such as ambient temperature, tilt or orientation of the instrument on the observed gravity values. To this end, several sensitivity tests have been performed with the instrument in an attempt to mimic to some extent changing conditions that one might be confronted with during field campaigns. Nevertheless, the value and impact of this paper could be enhanced if the study actually included real field tests to assess the capacities of a field instrument, as the title implies. Beyond the tests reported here under observatory or lab conditions,

several aspects that might be at least similarly relevant for field operation and impact the accuracy, precision and repeatability are missed, such as the impact of wind, humidity, of varying ambient temperatures and insolation along (laser) cables, vibrations and tilting of the laser unit itself, instabilities of power supply, frequent system mounting and un-mounting cycles, instruments displacements under warm conditions (without a full restart of the system at the new location), for instance.

Detailed comments:

Line 4-6: "... and hydrological mass changes". Syntax of sentence unclear, as it starts with "... instrument's performance in terms of ...".

Page 6, line 13: "... 1.2m height difference." Height difference relative to which reference?

Page 6, line 14: "In the Larzac observatory, estimated vertical gravity gradients on pillar 1 (FG5) and 2 (AQG) ...". On page 4, however it was mentioned that at Larzac the FG5 and the AQG were operated on the same pillar?

Page 7, line 18: "Small-scale repeatability was assessed using repeated gravity measurements on the same position in the gravity lab ...". Has this been carried out without moving and/or unmounting the instrument in between the measurements?

Page 8, line 5: "...horizontal atomic velocities are reduced and the AQG#B01 should not be sensitive to the Coriolis effect." But if there are remaining horizontal components (they are 'reduced' only) an impact of the Coriolis effect cannot be excluded.

Page 8, line 17: "... the iGrav#002 shows a higher sensitivity at short time scale, but an increase at long time due to environmental noise and tides residuals." The statement is not fully clear as these noise and tide effects can be expected to impact the iGrav and the AQG in a similar way? Furthermore, to which extent are the results presented in Figure 2 impacted by internal filtering of the iGrav data and thus not directly comparable to the AQG results?

[Figure]

Page 13, line 1: "Absolute comparison between both instruments [AQG and FG5] was not directly possible due to the set-up on different pillars and is impacted by the uncertainty related to the vertical gravity gradient (VGG) correction." But the others give an assessment of this VGG uncertainty so that in view of this uncertainty the absolute values of both instruments could in fact beshown and compared.

Page 13, line 7: "based on" instead of "based".

Page 13, line 10: What exactly has been done wit the instrument in between measurements on the same location? This might be relevant information for setting the reported repeatability in a context, also with respect to displacements during future field campaigns.

Page 13, line 16: "The measurement on point 2 was carried out using rubber pads under the tripod which added a height of 1.2 cm.". Why hasn't the same setup been used to assure full compatibility of the two measurements?"

Page 13, line 19: "To summarise, these first results show a repeatability better than 50 nm.s−2 . . ." Not clear where this value of 50 comes from given the results presented before.

---

## Referee Comment (RC2) · Anonymous Referee #2 · 13 Oct 2020

The paper presents the first results of analysis of performance of AQG#B01 quantum gravimeter that is proposed as a novel field instrument for high-frequency absolute gravity monitoring experiments. The performance of the instrument was characterized by three criteria: stability (absence of instrumental drift), sensitivity in relation to other gravimeters, and ability to react to hydrogeological mass changes. The accuracy of the gravity measurements by the instrument was compared to that of state-of the art absolute gravimeter (Micro-g-LaCoste, FG5#228). The study included a number of test measurements performed in the observatory conditions (Larzac Observatory in southern France). The results of the tests are carefully described and analysed in the paper. Generally, the study demonstrates that quantum gravimeter has the performance comparable to that of state-of-the art instrument, if installed in the ideal observatory

conditions. However, the purpose of the study was to demonstrate that the AQG#B01 quantum gravimeter can be used in demanding field conditions in geophysical and hydrological studies. This has not been demonstrated by the authors, and several open questions remain after reading the paper. There is reference to the earlier paper by Ménoret et al. (2018) describing the previous version of the instrument, but the authors were using the upgraded version and they did not provide many technical parameters of this upgraded instrument. Below is the list of some general questions that need to be addressed. 1) The instrument proposed for operation in the field conditions need to be compact and robust. It would be useful to present some general scheme of the instrument (or photo with general view of installation), from which it is possible to see its size. The authors provided only information that "improvement in ease of use and transportability has been achieved with each element weighing 40 kg or less". So the reader may be puzzled: how many elements weighting 40 kg is necessary to move from one site to another during field measurements? 2) One important property of any geophysical field instrument is autonomous power supply and its time of operation without recharging or changing the batteries. There is no information about this in the paper. The tests were done in observatory conditions, and there is no information about the instrument power supply system in the paper. What batteries are used to ensure the autonomous operation? Is the instrument using internal or external batteries? How long time the instrument can be operated autonomously without recharging or changing batteries? Is the instrument performance degrading with decreasing the battery capacity? 3) It was written in the Abstract that the proposed instrument can be used in high-frequency absolute gravity monitoring, but there is no information about signal bandwidth and dynamic range. It is not clear what the authors mean by high-frequency because the paper presents the data integrated over time intervals of 24 hours. 4) Each field instrument needs to have a system for data registration, either to high-capacity internal storage media or to remote storage using some data transmission protocols. There is no any information about this in the paper. 5) How the data quality control during measurements is realised? How much efforts from the operator

are necessary for this? 6) The thermal stability of the instrument was tested in a very narrow range from +20C to +30C. So the question arises: how the instrument performs in real conditions, in which it is possible to have daily temperature variations up to 20 C in some cases? Moreover, is it possible to use this instrument for monitoring mass changes in glaciated areas with temperatures lower than 0 C? 7) In their observatory tests the authors were using thermal insulation of the instrument, but realisation of the insulation is not discussed. It is important to discuss how the insulation was done, in order to demonstrate that the instrument can be insulated also in real field conditions. Is it possible to reach similar temperature stability as in the observatory? 8) The authors did not provide any reference to the manufacturer. Only name Muquans is mentioned, but it is not clear from the text whether it is a company name or the name of an individual.
* * *

---

## Author Comment (AC1) · 15 Oct 2020

We would like to thank both reviewers for their valuable comments. We agree with the two reviewers that additional factors should be included in the assessment of a field instrument, such as the impact of wind, humidity, cable insulation, and the further mentioned aspects. These are indeed interesting and important aspects and will eventually be evaluated, as the assessment of the instrument is still work in progress. Moreover, we understand from both reviews, that additional instrumental details and information on the measurement protocol need to be included in the manuscript. We will provide the missing information. Apart from these points, we do agree that a wider temperature range needs to be tested in the future.

[Figure]

One of the main points that has been made in both reviews is that this study focuses predominately on observatory conditions that have been altered to test specific aspects of field conditions. The chosen measurement conditions indeed do not resemble real terrain conditions in all possible aspects. This suggests that the title probably has to be adapted to make clear that we investigate 'semi-terrain' conditions or preparatory tests in an iterative approach.

Since the submission of the article, we conducted gravity measurements in a non-observatory, less controlled environment. We would like to briefly present the results here as they represent the next step towards deployment under field conditions and may answer some questions raised.

The measurements have been carried out in a garage located in an urban area in Montpellier, France. The garage is neither insulated nor air-conditioned. During two weeks, the temperature in the garage varied between 19 and 30 °C, due to the diurnal cycle. Moreover, short-term and rapid temperature changes of several degrees C have been achieved by opening the front and back garage doors in the early morning and evening. The doors have been left open for up to several hours. We did not explicitly measure wind or humidity changes. However, opening the doors clearly caused noticeable air circulation in the room. During those two weeks, both dry and sunny as well as very rainy weather conditions have been observed. Furthermore, the urban area showed increased vibrational noise levels. Under all these conditions, the AQG#B01 time series was stable and did not appear to be impacted by temperature or other changes.

---

## Editor Comment (EC1) · Ciro Apollonio (Editor) · 20 Nov 2020

In my opinion, further scrutiny by reviewers is needed, in order to verify the updated version. I kindly ask the reviewers for a quick response.

Kind regards

Ciro Apollonio
* * *

---

## Author Response (AR1)

**Author's response**

Anne-Karin Cooke, Cédric Champollion, Nicolas Le Moigne

19/11/2020

**Contents**

**Author's response to review # 1**

Dear Anonymous Referee # 1,

We would like to thank you again for your valuable comments. In the following, we would like to elaborate on how we included and responded to the points you raised. Our responses follow the order in which they are mentioned in your letter. Page and line numbers of the modified sentences refer to the updated version of the manuscript.

As a general statement, we propose to adjust the title to show in a more precise manner what we aimed at. In fact, the final field evaluation is a long-term undertaking. We do agree that many of the mentioned missing experiments are pertinent and we only show a selection of them in this paper. We consider these first tests to nevertheless provide useful and important information on crucial aspects of these new types of instruments. From our knowledge, this is the first publication presenting data and tests of a commercial quantum gravimeter by an academic research team.

We hope we could clarify unclear points and have responded adequately to your questions and concerns.

The new title of **"First evaluation of an absolute quantum gravimeter (AQG#B01) for future field experiments"** will be a more appropriate description of the scope of this paper.

We addressed your general and detailed comments in the updated version of the paper as shown below.

**1. Responses to general comments:**

*"Beyond the tests reported here under observatory or lab conditions,
several aspects that might be at least similarly relevant for field operation and impact
the accuracy, precision and repeatability are missed, such as the impact of wind, hu-*

*midity, of varying ambient temperatures and insolation along (laser) cables, vibrations and tilting of the laser unit itself, instabilities of power supply, frequent system mounting and un-mounting cycles, instruments displacements under warm conditions (without a full restart of the system at the new location), for instance."*

These are all valid points that need to be addressed in the further instrument assessment. For the impact of **wind, humidity, varying ambient temperatures and vibrations** we included results from measurements in a semi-terrain environment (garage), where we achieved a greater variability of these factors. As the current study is the first to be published, an exhaustive testing is out of the scope of the paper. Comparing to the numerous publications about for example the FG5, the current study points out some questions (azimuthal sensitivity) and provides first insights regarding the sensitivity of the AQGB#B to some main outdoor or field parameters such as external temperature, transient hydrological signals and displacement (between the Larzac Observatory, during the azimuthal test, and during the new experiment in the garage).

p. 18, l. 9

" The opening of the outward garage doors for a temperature change caused increased air circulation in the room. Possible impacts of these winds on tripod levelling and imbalance in temperature of different sides of the tripod cannot be ruled out and need to be investigated. We did not explicitly measure wind or humidity changes. However, opening the doors clearly caused noticeable air circulation in the room. During those two weeks, both dry and sunny as well as very rainy weather conditions have been observed without significant variations in the measured gravity by the AGQ."

We acknowledged the further aspects of **isolation along (laser) cables, vibrations and tilting of the laser unit itself, instabilities of power supply and frequent system mounting and unmounting cycles**
by addressing them in the discussion as follows:

p. 21, l. 19
"Further practical aspects such as the potential influence of cable insulation, unstable power supply or tilts of the electronic and laser systems are to be assessed in due time."

p. 21., l. 16
"In future experiments, the potential influence of strong winds needs to be assessed. Due to the very recent delivery of the instrument, displacements between observatory locations and system mounting and unmounting cycles have not been as frequent and need to be repeated."

We clarify that the mentioned aspect of **instruments displacements under warm conditions (without a full restart of the system at the new location),**
has to certain extend in fact already been assessed, and mentioned as:

p. 9, l. 20

"The repeatability is therefore assessed under warm conditions without a full restart of the system."

**2. Responses to detailed comments:**

- Line 4-6: ". . . and hydrological mass changes". Syntax of sentence unclear, as it starts with ". . . instrument's performance in terms of . . .".

p.1, line 6
"Furthermore, the measurements allowed for the successful monitoring of a hydrological mass change."
- Page 6, line 13: ". . . 1.2m height difference." Height difference relative to which reference?

p. 7, l. 19
"The vertical gravity gradient required to correct for the vertical gravity differences between the FG5#228 and AQG#B01 measurement locations in Larzac and Montpellier have been estimated with a relative Scintrex CG5 and CG6 gravimeter. The vertical gravity gradients were estimated from measurements on two heights of 1.2 m vertical difference. "

- Page 6, line 14: "In the Larzac observatory, estimated vertical gravity gradients on pillar 1 (FG5) and 2 (AQG) . . .". On page 4, however it was mentioned that at Larzac the FG5 and the AQG were operated on the same pillar?

p. 4, l. 5
" In the Larzac observatory, the AQG#B01 and FG5#228 were operated on the same concrete pillar with approximately one meter distance between both instruments. "

p. 7, l 22
"In the Larzac observatory estimated vertical gravity gradients on measurement point p1 (FG5) and p2 (AQG) on the same pillar are -3.225 and -3.220 kE, respectively, averaged over one year of monthly measurements (Cooke et al., in preparation)."

p.14, l. 13

"Absolute comparison between both instruments is limited due to the set-up on different locations on the pillar[...]"

- Page 7, line 18: "Small-scale repeatability was assessed using repeated gravity measurements on the same position in the gravity lab . . .". Has this been carried out without moving and/or unmounting the instrument in between the measurements?

p. 9, l. 19

"In between these measurements, the instrument was not turned off or disassembled, only the measurement was temporally stopped and the sensor head was moved within the room."

- Page 8, line 5: "...horizontal atomic velocities are reduced and the AQG#B01 should not be sensitive to the Coriolis effect." But if there are remaining horizontal components (they are 'reduced' only) an impact of the Coriolis effect cannot be excluded.

p. 9, l. 26

"By symmetrical construction (hollow pyramidal reflector and location of the detection photo-diodes), horizontal atomic velocities are reduced and the AQG#B01 should only show a negligible sensitivity to the Coriolis effect as in the CAG gravimeter (Louchet-Chauvet et al., 2011)."

Louchet-Chauvet, A., Farah, T., Bodart, Q., Clairon, A., Landragin, A., Merlet, S., & Dos Santos, F. P. (2011). The influence of transverse motion within an atomic gravimeter. *New Journal of Physics*, *13*(6), 065025.

- Page 8, line 17: "... the iGrav#002 shows a higher sensitivity at short time scale, but an increase at long time due to environmental noise and tides residuals." The statement is not fully clear as these noise and tide effects can be expected to impact the iGrav and the AQG in a similar way? Furthermore, to which extent are the results presented in Figure 2 impacted by internal filtering of the iGrav data and thus not directly comparable to the AQG results?

Indeed, the tidal effects are expected to impact the instruments in a similar way, as clarified hereafter:

p. 10, l. 10

"All three instruments show a slight increase at long time likely due to environmental noise and tides residuals."

Concerning the filtering of the iGrav data, no special processing other than classical averaging has been performed, thus we do not expect any inconsistencies here. A small effect of dephasing could be due to a different time lag of the instrumental response which is small for the iGrav. A large time lag in the instrumental response of the AQG#B would have been seen during the tide analyses. In a future experiment, one should estimate the instrumental response of the AQG#B.

- Page 13, line 1: "Absolute comparison between both instruments [AQG and FG5] was not directly possible due to the set-up on different pillars and is impacted by the uncertainty related to the vertical gravity gradient (VGG) correction." But the others give an assessment of this VGG uncertainty so that in view of this uncertainty the absolute values of both instruments could in fact be shown and compared.

This is true for the Montpellier site (Géosciences), for the Larzac site the VGG correction might not be constant over time due to hydrological effects.

The sentence was adjusted :
p.14, l. 13

"Absolute comparison between both instruments is limited due to the set-up on different pillars and is impacted by the uncertainty related to the vertical gravity gradient (VGG) correction."

p. 14, l. 20

"Likely due to soil moisture changes, VGG at the Larzac site may vary by several tens of E over weeks and months (Cooke et al., in preparation). We recommend to take the accuracy assessment at the Larzac site with caution and to repeat the measurements together with the FG5 in the near future."

- Page 13, line 10: What exactly has been done wit the instrument in between measurements on the same location? This might be relevant information for setting the reported repeatability in a context, also with respect to displacements during future field campaigns.

p. 9, l. 19

"In between these measurements, the instrument was not turned off or disassembled, only the measurement was temporally stopped and the sensor head was moved within the room."

- Page 13, line 16: "The measurement on point 2 was carried out using rubber pads under the tripod which added a height of 1.2 cm.". Why hasn't the same setup been used to assure full compatibility of the two measurements?"

As the comparison between the two measurement locations in the same room in the Géosciences laboratory with the AQG#B01 has been carried using slightly different set-ups to test the impact of the rubber pads. No significant effect have been seen in the AQG#B averaged gravity measurements except at very small small time scale (<10 min) where dispersion is reduced by the rubber pads. We agree that this information can be misleading and is in fact irrelevant concerning the repeatability assessment. We therefore changed the following paragraph:

"AQG#B01 was operated on two measurement points within the same room in the Géosciences laboratory at about one meter distance, of which one serves FG5#228 measurements. At this short distance, no considerable horizontal difference in g is expected. CG6 relative gravity measurements were carried out with two instruments (CG6#120 and CG6#125) on 26/03/2020 on both points and found a negligible difference in g of $2 \pm 6$ nm.s$^{-2}$. The difference in g measured with the AQG#B01 between point 1 and point 2 is $15 \pm 48$ nm.s$^{-2.}$
The measurement on point 2 was carried out using rubber pads under the tripod . The rubber pads has not significant effect in the 10 min averaged gravity readings. Only a small

decrease of the gravity readings dispersion at short time scale (< 10 minutes) was observed."

- Page 13, line 19: "To summarise, these first results show a repeatability better than 50 nm.s−2 . . ." Not clear where this value of 50 comes from given the results presented before.

p. 15, l. 11

"To summarise, these first results show a repeatability of 3 nm.s$^{-2}$ with a standard deviation of 25 nm.s$^{-2}$ and no detectable drift over 2 months operation."

**Author's response to review # 2**

Dear Anonymous Referee # 2,

We would like to thank you again for your valuable comments. In the following, we would like to elaborate on how we included and responded to the points you raised. Our responses follow the order in which they are mentioned in your letter. Page and line numbers of the modified sentences refer to the updated version of the manuscript.

As a general statement, we propose to adjust the title to show in a more precise manner what we aimed at. In fact, the final field evaluation is a long-term undertaking. We do agree that many of the mentioned missing experiments are pertinent and we only show a selection of them in this paper. We consider these first tests to nevertheless provide useful and important information on crucial aspects of these new types of instruments. From our knowledge, this is the first publication presenting data and tests of a commercial quantum gravimeter by an academic research team.

We hope we could clarify unclear points and have responded adequately to your questions and concerns.

The new title of **"First evaluation of an absolute quantum gravimeter (AQG#B01) for future field experiments"** will be a more appropriate description of the scope of this paper.

We addressed your general and detailed comments in the updated version of the paper as shown below.

**Responses to comments:**

- 1) The instrument proposed for operation in the field conditions need to be compact and robust. It would be useful to present some general scheme of the instrument (or photo with general view of installation), from which it is possible to see its size.

A photo of the instrument and all necessary units is given for a more detailed illustration of its size.

- The authors provided only information that "improvement in ease of use and transportability has been achieved with each element weighing 40 kg or less". So the reader may be puzzled: how many elements weighting 40 kg is necessary to move from one site to another during field measurements?

p.4, l.14

"The sensor head, the laser system and the control unit come each with a dedicated transport box that can be carried by two people. A fourth transport box is provided for the tripod, cables, connectors and the laptop."

- 2) One important property of any geophysical field instrument is autonomous power supply and its time of operation without recharging or changing the batteries. There is no information about this in the paper. The tests were done in observatory conditions, and there is no information about the instrument power supply system in the paper. What batteries are used to ensure the autonomous operation? Is the instrument using internal or external batteries? How long time the instrument can be operated autonomously without recharging or changing batteries? Is the instrument performance degrading with decreasing the battery capacity?

The raised questions are important but are out of the scope of the paper. They will be under evaluation once the sensitivity to all environmental parameters has been extensively studied. In the course of such a validation, all potential biases or noise sources due to external parameters need to be determined first to clearly estimate the power supply impact. This aspect is clearly one of the first points on our agenda. Power consumption is 250 W, as can be found here: https://www.muquans.com/product/absolute-quantum-gravimeter/.

p. 4, l. 12

"Battery operation is possible but has not been tested yet. This study focuses on operation relying on external power supply."

- 3) It was written in the Abstract that the proposed instrument can be used in high-frequency absolute gravity monitoring, but there is no information about signal bandwidth and dynamic range. It is not clear what the authors mean by high-frequency because the paper presents the data integrated over time intervals of 24 hours.

The term "high-frequency" has been deleted in the abstract for clarity. High-frequency is relative to the signal or the process studied. The 2 Hz sampling rate is indeed mentioned later in the manuscript and the raw gravity files provided by the AQG#B are given at 2 Hz.

The corrected gravity data are averaged over 10 minutes. We show the gravity data as 24 h averages due to the best sensitivity at this integration interval.

- 4) Each field instrument needs to have a system for data registration, either to high-capacity internal storage media or to remote storage using some data transmission protocols. There is no any information about this in the paper.

p. 4, l. 18

"The AQG GUI software allows for measurement control, data storage and processing on a connected laptop."

p. 7, l. 1

"For the AQG#B01, the mentioned corrections are carried out using the AQG GUI software."

- 5) How the data quality control during measurements is realised? How much efforts from the operator are necessary for this?

p. 4, l. 19.

"Calibrations prior to measurement start can be launched manually or automatically. Data quality control is provided by the software, re-calibration is initiated if predefined thresholds are undershot (e.g. number of atoms)."

- 6) The thermal stability of the instrument was tested in a very narrow range from +20C to +30C. So the question arises: how the instrument performs in real conditions, in which it is possible to have daily temperature variations up to 20C in some cases? Moreover, is it possible to use this instrument for monitoring mass changes in glaciated areas with temperatures lower than 0 C?

We acknowledge and address the necessity of experiments in a larger temperature range in the conclusions. The results shown and discussed here relate only to the temperature range from the AQG#B datasheet available on the Muquans website (https://www.muquans.com/product/absolute-quantum-gravimeter/). To our knowledge, at low temperatures, reaching a sufficient number of atoms in the vacuum chamber poses a challenge for gravity measurements.  As from experience with the FG5, we believe that a tent or equivalent shelter should be provided for field experiments at such a low temperature.

With the additional information given by the newly added paragraph on measurements in a semi-terrain environment (garage), we show that rapid temperature changes did not influence the measurement of g.

"Operation under semi-terrain conditions" p. 9 first paragraph and p. 18 first paragraph

- 7) In their observatory tests the authors were using thermal insulation of the instrument, but realisation of the insulation is not discussed. It is important to discuss how the insulation was done, in order to demonstrate that the instrument can be insulated also in real field conditions. Is it possible to reach similar temperature stability as in the observatory?

p. 16, l.9

"The sensor head can be easily wrapped in the fitting insulation (down) cover."

The isolation of the sensor head with the provided down cover is not required for the tested temperature range (20 – 30°C) and has not been used in the assessment of this range. It has been shown to be useful for instrument starts below 20 °C, however, this is still under investigation.

- 8) The authors did not provide any reference to the manufacturer. Only name Muquans is mentioned, but it is not clear from the text whether it is a company name or the name of an individual.

[revised manuscript text omitted]

---

## Referee Report (RR1)

**Manuscript gi-2020-22**
**Reviewer #3 comments and suggestions.**

As this is the second round of review, there isn't much I could find to point out. As to its content the paper can be published after considering the bubbles in the pdf and comments below, leaving the response to the authors. Some clerical issues will probably be addressed in the pre-publishing correspondence with the GI office; nevertheless I shall point them out in the last section below. As you can guess, I don't request anonymity.

**Major points:**
Page 6, line 15ff: If possible, specify the uncertainty of the observed gradient values.

Page 14, line 5: The bias due to tilt $\zeta$ goes with $g$ (1-cos $\zeta$). Say " and---at small angles---increases in a quadratic way..."

Page 14, line 18: No effect from precipitation on iGrav? Related to its location in the lab? Any obvious/plausible reason? Perhaps insert "for no obvious reason".

Page 17, lines 18-20: Depends on how you respond to the previous point. I wonder how the effect escaped capture in the iGrav. The conclusive character of the remark on line 20 appears scantly underpinned to me. Yet, acknowledging the problems with drift in SG's, the expected (or proven) long-term stability of an AQG will be is primary asset.

Page 18, cautioning against an overreach, the performance of the AQG in comparison with an OSG as concerns noise and resolution of sub-daily to weekly phenomena on multi-year time scales (e.g. Scherneck et al., 2020; papers by van Camp and many others) still needs evidence. For the applications mentioned, this is not crucial, however.

Figure 6. While those figures with short ordinate axes don't allow for axis titles, for this one I propose to add titles "Dates in year 2020" and "Gravity deviation [nm s$^{-2}$]".

Figures 2 and 3, titles "Lag $\tau$ [s]" and "Allen deviation [nm s$^{-2}$]". Figure 2 has awkwardly formatted numbers along the abscissa, o.k. in Fig. 3. "$10^2 \ 10^3$ …" to be preferred; or "2 3 ..." and title "Lag [log $\tau$/s]" if superscripts turn out illegible; or scale to kilo-seconds [ks], but that's unusual.

For future work into comparing different gravimeters, published papers that I authored might be helpful. In short:
   The two FG5's I analysed for drop scatter showed (a) an over-compensation of ground vibarition in the frequency range above 0.1 Hz; (b) an almost perfect retrieval of seismometer-like acceleration at typical surface-wave frequencies (> 30 s periods).
   The OSG (GWR_054) I analysed has the so-called GGP low-pass filter, which distorts the signal at seismic frequencies (from 100 s periods and shorter). A broadband seismometer proved useful to retrieve the ground acceleration in this range. For the tranfer spectra between different sensors a numerical equivalent of the GGP-filter was used, the OSG serving as a reference.
   Thus, you could improve the evaluation of different means to isolate the AQG for ground vibarations. A number of FG5 setups were shown to be affected by drifts, much larger than what could be blamed on the OSG.
   See Scherneck and Rajner (2019); Scherneck at al., (2020).

**Technical issues.**
   • Units should appear in upright style, foremost nm s$^{-2}$ (no dot!) and μrad.

- In the text, e.g. "Residuals of g refer to the mean of the shown measurements, respectively" (Caption Fig. 4), variables should appear in slanted style, *g*. It already does at most places.
- Integers up to twelve (some say ten) should be spelled out, e.g. two instead of 2, unless a unit follows. The resolve is pretty ambiguous (https://www.englishgrammar.org/spell-numbers/, https://www.chicagomanualofstyle.org/qanda/data/faq/topics/Numbers.html?page=1)
- For dates, ISO 8601 is to be preferred, e.g. 2020-02-19 for 19/02/2020, alternatively following British English (Feb. 2, 2020).
- Unless Karst is the name of a specific geologic body, uncapitalise it (karst).

**References**

Scherneck, HG., Rajner, M. (2019), Using a superconducting gravimeter in support of absolute gravity campaigning—a feasibility study. Geophysica 54(1):117–135(2019), https://eartharxiv.org/repository/view/916/

[revised manuscript text omitted]

---

## Author Response (AR2)

**Authors' response to reviewers 3 and 4**

Anne-Karin Cooke, Cédric Champollion, Nicolas Le Moigne

15/01/2021

**Authors' response to review # 3**

Dear Hans-Georg Scherneck,

we thank you for your valuable comments and suggestions on how to improve the manuscript. We included all comments, adjusted the figures and corrected the mentioned technical aspects. The corrections are marked in the pdf showing the differences.

Concerning these remarks hereafter we respond in the following way:

- Page 6, line 15ff: If possible, specify the uncertainty of the observed gradient values.

p.7 , l. 22
"In the Larzac observatory estimated vertical gravity gradient on measurement point p1 (FG5) is -3.226 kE with a standard deviation of 0.022 kE and -3.220 kE with a standard deviation of 0.017 kE for measurement point p2 (AQG), respectively, averaged over one year of monthly measurements (Cooke et al., in preparation)."

- Page 14, line 5: The bias due to tilt ζ goes with g (1-cos ζ). Say " and---at small angles---increases in a quadratic way…"

p. 15, l. 7
"A tilt bias of 50 μrad leads to an error of 12.3 nm s$^{-2}$ and at small angles increases in a quadratic way which leads to 49 nm s$^{-2}$ for 100 μrad."

- Page 14, line 18: No effect from precipitation on iGrav? Related to its location in the lab? Any obvious/plausible reason? Perhaps insert "for no obvious reason".
- Page 17, lines 18-20: Depends on how you respond to the previous point. I wonder how the effect escaped capture in the iGrav. The conclusive character of the remark on line 20 appears scantly underpinned to me. Yet, acknowledging the problems with drift in SG's, the expected (or proven) long-term stability of an AQG will be is primary asset.

Our excuses that these lines have been unclear. In fact, there was not much precipitation. There was no effect that the iGrav would have missed.
p.15, l.22
"Precipitation was negligible during these measurements and no significant effect of precipitation was observed in the iGrav#002 monitoring. "

- Page 18, cautioning against an overreach, the performance of the AQG in comparison with an OSG as concerns noise and resolution of sub-daily to weekly phenomena on multi-year time scales (e.g. Scherneck et al., 2020; papers by van Camp and many others) still needs evidence. For the applications mentioned, this is not crucial, however.

p. 20, l. 11

"On multi-year time scales the performance of the AQG in comparison with superconducting gravimeters still requires evidence with respect to noise and resolution of sub-daily to weekly phenomena (Scherneck et al., 2020)."

We would like to thank Hans-Georg Scherneck for his advice and suggestions on important aspects and the suggested references.

**Authors' response to review # 4**

Dear anonymous referee,

we thank you for your corrections and suggestions on how to improve the manuscript. We agree with your comments and included them as well as restructured the abstract.

- The changes in the title of the manuscript are partially satisfactory with respect to some concerns raised by previous reviewers. The work now right presents itself as a "first evaluation" of an experiment whose validity must be deeply verified. However, the Abstract should be rewritten in a clearer, more concise way and without repetition.

The abstract has been modified and rewritten in a more structured way.

- The various additions made partially fill the gaps in the initial version of the manuscript. However, some steps are treated rather roughly. I am referring in particular to the experimentation done in a semi-terrain environment (garage) and to the problems relating to the various physical variables (wind, humidity, temperatures, vibrations, ...) that may critically affect the experiment itself. I believe that the work made by the authors deserves to be published but after further revisions and technical corrections, and as a "Technical Report". However, the authors are advised to define the next steps for the full validation of the experiment.

From a metrological point of view, it is not possible to achieve a field measurement with an exhaustive error budget as all the environment parameters and their variability can not be

measured in a practical field experiment. The referee is certainly right about this statement. From the point of view of a field geophysicist, measurements have to be obtained and the various physical variables are indeed "roughly" treated. However, first order effects of the physical variables can be obtained from literature to get a first estimate of their importance. During the garage experiment, the gravity value should not change as no rainfall was recorded. If the AQG provides a 7-days gravity dataset with almost no gravity readings changes greater than 20 nms$^{-2}$, one can estimate that the various physical parameters (wind, humidity, vibrations etc.) at the first order are not affecting the measurements. This is a first important step and indeed one has to go further.

- I point out that the additional sentence "without significant variations in the measured gravity by the AGQ" (as response to a comment of the reviewer 1) does not appear in the revised text (cf. p. 19, line 2), and this is not the only case. In the amended text, I did not find even the sentence: "In future experiments, the potential influence of strong winds needs to be assessed. Due to the very recent delivery of the instrument, displacements between observatory locations and system mounting and unmounting cycles have not been as frequent and need to be repeated". Note that the last sentence is part of the response to a comment from the first reviewer and should fill some serious experimental gaps.

The missing sentences from the reply letter have been inserted into the manuscript.